# Controlling Nutritional Status (CONUT) Score and Prognostic Nutritional Index (PNI) Are Good Candidates for Prognostic Markers for Acute Pancreatitis

**DOI:** 10.3390/medicina59010070

**Published:** 2022-12-28

**Authors:** Mustafa Zanyar Akkuzu, Engin Altıntaş, Serkan Yaraş, Orhan Sezgin, Fehmi Ateş, Enver Üçbilek, Osman Özdoğan

**Affiliations:** Department of Gastroenterology, Mersin University Faculty of Medicine, Mersin 33343, Turkey

**Keywords:** acute pancreatitis, CONUT score, PNI, nutrition, severity, prognosis

## Abstract

*Background and Objectives*: It has been demonstrated that parameters such as the Controlled Nutrition Status (CONUT) score and Prognostic Nutrition Index (PNI) are beneficial for the assessment of patients’ nutrition. In this study, our objective was to investigate the potential benefits of CONUT and, as a prognostic marker of acute pancreatitis, the PNI. *Materials and Methods*: The data of 361 patients were analysed retrospectively. The PNI and CONUT scores of these patients were retrospectively calculated. They were categorised as CONUT-high (≥3) and CONUT-low (≤2). A PNI ≥ 45 was considered high and a PNI < 45 low. The AP severity and organ failure due to disease were evaluated based on Atlanta 2012. *Results:* According to the CONUT score, it was found that 209 patients had normal to mild, whereas 152 patients had severe malnutrition. A total of 293 patients had mild AP and 68 thereof had severe AP. The patients with a high CONUT score used more antibiotics, were hospitalised more in intensive care units and experienced organ failure more frequently. There were no intensive care hospitalisations, mortalities, surgical needs and local complications among the patients with a higher PNI score. *Conclusions*: CONUT and the PNI have proven to be useful prognostic markers not only for predicting nutritional status but also for estimating the severity and results of AP.

## 1. Introduction

Acute pancreatitis is a non-bacterial acute inflammation of the pancreas, which occurs due to the gland’s auto-digestion caused by the entry of pancreas enzymes into the parenchyma and the activation thereof based on etiological factors, and which can be clinically and histologically regressed. It is an acute picture varying from a self-limiting mild pancreatic inflammation to life-threatening advanced systemic symptoms, characterised by, as laboratory signs, high levels of pancreatic enzymes in the blood and urine and most often by a stomach ache in a physical examination. Acute pancreatitis manifests itself through frequently repeated episodes. The recurrence of episodes may cause permanent damage to the pancreas, and as a result, chronic pancreatitis or pancreatic failure may ensue. Acute pancreatitis does not have a specific treatment and it may be mortal or morbid in its progression [1,2,3,4,5,6].

A high amount of catabolic stress arises as a result of a hyper-dynamic and systemic inflammatory response syndrome accompanied by a hypermetabolic status characterised by acute pancreatitis-induced increased protein catabolism, lipolysis as well as glucose intolerance [7]. In these patients, protein catabolism rises by 80% and an energy deficit by 20% [8].

In acute pancreatitis, nutritional support varies depending on the severity of the disease. Mild to moderate pancreatitis does not have much effect on nutritional status and metabolism and patients can generally return to their normal nutrition within 3–7 days. In severe AP, the protein energy deficit and protein catabolism increase, and there may be a negative nitrogen balance, which can engender a negative impact on both the nutritional status and the progression of the disease [9]. A study showed that the mortality rates of patients with a negative nitrogen balance were 10 times higher as compared with those having a normal nitrogen balance [10].

Recently, markers of inflammatory and nutritional status as measured by parameters such as the Controlled Nutritional Status (CONUT) score and the Prognostic Nutritional Index (PNI) were shown to be successful in predicting a poor prognosis and postoperative complications in various cancer patients receiving chemotherapy or undergoing surgical resection. These inflammatory and nutritional status parameters rely on serum and/or blood counts in peripheral blood and are routinely measured in daily clinical practice [11,12,13,14,15]. In this study, we investigated the potential use of the PNI and CONUT as prognostic markers in acute pancreatitis (AP).

## 2. Materials and Method

Hospital files of all patients who were followed up with a diagnosis of AP in our gastroenterology clinic over the last 7 years were retrospectively reviewed. A total of 361 patients with a confirmed diagnosis of acute pancreatitis were included in the study. Approval was obtained for this study from Ethics Committee of the Mersin University (Date: 30 October 2021; Issue No: 1867977). Demographic data and risk factors were recorded for each patient. The scores of the patients calculated according to the AP severity scoring systems were obtained from file reviewing and recorded for each patient. The diagnosis of acute pancreatitis was made by having 2 of the criteria of typical abdominal pain, amylase and lipase more than 3 times normal and presence of imaging findings. Patients under 18 years of age, patients previously diagnosed with chronic pancreatitis, patients with malignancies and multiple organ failure were excluded from the study. Patients were divided into two groups as mild and severe according to the Atlanta criteria. Patients in the mild pancreatitis group whose signs of organ failure improved within the first 48 h were included. Patients with organ failure and complications lasting longer than 48 h were included in the severe pancreatitis group. The APACHE-2 scores of the patients hospitalised in the intensive care unit were calculated. Cases below 8 points were accepted as mild pancreatitis and cases above 8 points as severe pancreatitis [7,16,17].

CONUT score and PNI were calculated retrospectively according to the laboratory values of the patients when they were first diagnosed. The CONUT score and PNI consist of different laboratory indicators. Using the PNI albumin and lymphocyte count, the CONUT score is calculated based on the albumin, lymphocyte count and total cholesterol values.

PNI is an index calculated from serum albumin and peripheral blood lymphocyte count used to assess the immunological and nutritional status in people with digestive diseases. It is not a routine scoring used in the severity scoring of AP. The Prognostic Nutritional Index was used successfully in patients with acute pancreatitis, and it was found that the mortality rate was higher if the PNI value measured at the beginning of the disease was lower for 100-day mortality [18,19,20]. The result obtained by adding 10 times the serum albumin level and 5 times the number of lymphocytes [(10xalbumin) + (lymphocytesx5)] was recorded for each patient. Patients with a PNI score greater than 45 were considered at normal risk, and patients with a PNI score less than 45 were considered at risk of severe malnutrition.

The Control of Nutritional Status Score (CONUT), a screening tool that assesses nutritional status using patients’ biochemical findings, is practical and easy to use in hospitalised patients. It is calculated with albumin, lymphocyte and cholesterol values [21] (Table 1). Albumin represents protein reserves, total cholesterol represents calorie depletion and lymphocyte count represents immune defence. A decrease in each component is assigned a high score. Therefore, a high score means worse nutritional status [22]. This scale was used as an indicator of severity and mortality in many pathological conditions, particularly heart conditions and gastrointestinal tumours [23,24,25,26,27]. Patients were categorised as CONUT-high (≥3) and CONUT-low (≤2) according to their CONUT scores.

Antibiotic use was defined as antibiotics administered to patients with acute cholecystitis or infected necrosis. The operation was performed in patients with necrotising pancreatitis. Necrosectomy and abscess drainage were performed on the patients.

## 3. Statistical Analysis

Variables were expressed as frequency, mean and standard deviation. Patients were categorised according to CONUT score (low ≤ 2-high ≥ 3), Revised Atlanta criteria (mild-severe) and PNI scores (PNI ≥ 45 and PNI < 45). Pearson Chi-square test was used to investigate whether there was a difference between the subgroups in terms of CONUT score, Revised Atlanta criteria and PNI scores. For continuous variables, Student’s *t*-test was used to investigate the difference between groups if normal distribution was observed. Mann–Whitney U test was used for parameters outside normal distribution. For statistical analysis, IBM SPSS Statistics, Version 24 (IBM, A.B.D), package program was used. *p* ≤ 0.05 was considered statistically significant.

## 4. Results

A total of 361 patients were included in the study as a total sum. Of the total 361 patients, 184 (51%) were female and 177 (49%) were male, with a mean age of 54.8 ± 17 years. The mean age of the women was 54.6 ± 6 years, whereas the mean age of the men was 55 ± 17 years. No significant difference was found between the two sexes in terms of age. According to the aetiology of AP, 222 (61.5%) were biliary, 33 (9.1%) were alcoholism, 24 (6.6%) were hypertriglyceridemia and 82 (22.7%) were other etiological factors (tumour, post ERCP, oddi fibrosis, etc.) (Table 2).

Of the patients, 57.9% had a low CONUT score and 42.1% had a high CONUT score. From among the patients with high CONUT scores, 25% had severe pancreatitis according to the Atlanta classification. In 14.3% of the patients with a low CONUT score, severe pancreatitis was present (*p* < 0.05). Similarly, 80.4% of the patients with a low CONUT score and 75.6% of the patients with a high CONUT score had a high APACHE score (Table 2). In other words, there was a statistically significant relationship between the CONUT score and the severity of pancreatitis (*p* < 0.05).

Surgery was needed in 4.6% of the patients with a high CONUT score and 5.2% of the patients with a low CONUT score. There were local complications in 11.8% of the patients with a high CONUT score and 7.17% of the patients with a low CONUT score (Table 2). The CONUT score was not significantly associated with the need for surgery (*p* = 0.489) and local complications (*p* = 0.092).

Antibiotic use was defined as antibiotics administered to patients with acute cholecystitis or infected necrosis. Ceftriaxone was started in patients with concomitant cholecystitis and imipenem in patients with infected necrosis. Antibiotics were needed in 42.1% of the patients with a high CONUT score and 27.2% of the patients with a low CONUT score (Table 3). There was a statistically significant correlation between the CONUT score and the need for antibiotics (*p* < 0.05).

The mean hospitalisation duration of the patients with a high CONUT score was 6.4 ± 4.7 days, while the mean hospitalisation duration of the patients with a low CONUT score was 4.2 ± 3.5 days. A total of 9.8% of the patients with a high CONUT score were hospitalised in an intensive care unit, while 4.3% of the patients with a low CONUT score were hospitalised in an intensive care unit. Organ failure developed in 34.9% of the patients with a high CONUT score, while 23.9% of the patients with a low CONUT score suffered from it. Exitus occurred in 3.9% of the patients with a high CONUT score and 0.95% of the patients with a low CONUT score (Table 3). A high CONUT score had a significant relationship with the length of the hospital stay, the length of the stay in an intensive care unit, organ failure and death (*p* < 0.05).

When we analysed the PNI ≥ 45 and PNI < 45, 1.32% of the patients with a high PNI had severe pancreatitis, while 31.4% of the patients with a low PNI had severe pancreatitis. Whereas 15.9% of the patients with a low PNI were hospitalised in an intensive care unit, none of the patients with a high PNI were hospitalised in an intensive care unit. The PNI scores of all the patients were low. Whereas the mean hospitalisation duration of the patients with a low PNI score was 5.04 ± 3.8, the mean hospitalisation duration of the patients with a high PNI score was 4.72 ± 3.14. A total of 48% of the patients with a low PNI needed antibiotics, while 13.2% of those with a high PNI required antibiotics. Whereas 11.9% of the patients with a low PNI score needed surgery, none of the patients with a high PNI score required surgery. A total of 15.7% of the patients with a low PNI had local complications, while none of the patients with a high PNI had local complications. In other words, the PNI score was low in all (100%) of the patients with an intensive care hospitalisation, death, a need for surgery and local complications (Table 4). A low PNI was statistically associated with the severity of pancreatitis, death, the need for antibiotics, the length of the stay in an intensive care unit, the length of the hospital stay, the need for surgery and local complications (*p* < 0.05).

When we analysed the PNI ≥ 45 and PNI < 45, a total of 58.1% of the patients had a low PNI and 41.8% had a high PNI. The PNI was low in 61.9% of the patients with high CONUT scores and 38.1% of the patients with low CONUT scores. The PNI was high in 14.6% of the patients with a high CONUT score and in 85.4% of the patients with a low CONUT score (Table 5).

## 5. Discussion

Patients with acute pancreatitis (AP), especially severe cases, require appropriate nutritional management due to prolonged food restriction and increased caloric requirements. However, little is known about the relationship between the clinical status and nutritional status in patients with AP. Malnutrition due to disease is a common condition among hospitalised medical patients, with a prevalence of 20 to 50% [25,26,27]. The Early Nutritional Support to Frailty, Functional Outcomes, and Recovery of Malnourished Medical Inpatients Trial (EFFORT) has shown that the early commencement of individualised nutritional support reduced the complications and mortality in medical inpatients at risk of malnutrition [28,29,30,31]. Surprisingly, in this trial, there was little evidence on the related subgroup effects of nutritional status and type of disease. In our patients, it was found that acute pancreatitis had a better progression in patients with a high PNI score and a low CONUT score, i.e., those having a better nutritional status. Nevertheless, the inflammatory status of the patients may affect their responses to the nutritional support for various reasons without regard to the medical condition involved.

Inflammation has various metabolic effects, including increased insulin resistance and a decreased appetite, that cause the inhibition of nutrients entering the cells [32,33]. In fact, it is thought that regardless of the underlying disease, the inflammation is a remarkable driving force for disease-related anorexia, decreased food intake and muscle catabolism. This may also in part account for the lower results of patients in connection with the inflammation, including longer hospital stays and increased mortality [34,35]. The importance of inflammation in the pathogenesis of malnutrition is also reflected in the classification carried out by the European Society for Clinical Nutrition and Metabolism (ESPEN). They suggest further classifying malnutrition as related with inflammatory and non-inflammatory disease [36]. Inflammatory malnutrition associated with a disease is defined as a subsequent food intake deficiency or intake with a negative nutrient balance as well as underlying diseases causing inflammation [37].

In our study, in patients with high CONUT scores and a low PNI (i.e., those with a poor nutritional status), their pancreatitis severity, mean length of hospital stay, antibiotic needs, organ failure, requirements for intensive care, length of hospital stay and exitus rates were significantly higher. While the CONUT score was insignificant in terms of local complications and the need for surgery, the PNI score was significant. A total of 61.9% of those with a high CONUT score already had a low PNI score. This indicated a good correlation of both scores (*p* < 0.05).

Many studies have shown that malnutrition can increase inflammation and thus delay the healing of diseases. It was also shown that the CONUT score and PNI can predict mortality in many inflammatory diseases [38,39,40,41]. We are of the opinion that the underlying mechanisms not only closely relate to nutrition but also the acute exacerbation of the disease. We suggest that the CONUT score, a complex of immunity status, protein reserve and lipid metabolism, and the PNI score have an incrementally important impact on patients with AP regarding the assessment of the severity and prognosis. In this study, a higher CONUT score was found to be significantly associated with a lower PNI score, a poor inflammatory and nutritional status as well as severe AP. Therefore, in acute pancreatitis, we think that nutrition should be started as soon as the patient’s pain is relieved. We think that early and adequate nutrition can affect the prognosis very well in acute pancreatitis.

## 6. Study Limitations

This study is limited by its retrospective, uncontrolled design and small sample size. However, to our knowledge, this is the first study worldwide to demonstrate the utility of the CONUT score and PNI as a nutritional screening method and a predictor of severity in AP patients. These findings should be evaluated in further prospective studies.

## 7. Conclusions

These data showed that a nutritional assessment with the PNI and CONUT score was useful in predicting the prognosis of patients with AP. Therefore, we think that an appropriate intervention that improves the nutritional status of patients with high CONUT scores and a low PNI in acute pancreatitis may contribute to the improvement in the prognosis of patients with AP. In conclusion, the CONUT score and PNI score are correlated to assess the nutritional status in AP patients. We think that early nutrition therapy in FP can positively affect the prognosis by calculating the CONUT score and PNI easily.

## Figures and Tables

**Table 1 medicina-59-00070-t001:** Evaluation of nutritional status by way of Controlled Nutritional Status (CONUT) score.

Serum albumin (gr/mL) Score	≥3.5 0	3.0–3.4 2	2.5–2.9 4	<2.5 6
Total lymphocyte rate Score	≥1600 0	1200–1599 1	800–1199 2	<800 3
Total amount of cholesterol (mg/dl)	≥180 0	140–179 1	100–139 2	<100 3
Total Score	≤2 low		≥3 high	

**Table 2 medicina-59-00070-t002:** Patient characteristics.

Age	54.8 ± 17	
Gender	Female Male	51% 49%
Etiyology	-Gallstone -Alcoholism -Hypertriglyceridemia -Tumor, post ercp, oddi fibrosis, etc.	61% 9.1% 6.6% 22.7%

**Table 3 medicina-59-00070-t003:** Categorisations according to Controlled Nutritional Status (CONUT) score.

	High-CONUT ≥ 3	Low-CONUT ≤ 2	*p* Value
Severe pancreatitis	38 (25%)	30 (14.3%)	*p* < 0.001
Hospitalisation in intensive care unit	15 (9.8%)	9 (4.3%)	*p* = 0.02
Exitus	6 (3.9%)	2 (0.95%)	*p* < 0.001
Need for surgery	7 (4.6%)	11 (5.2%)	*p* = 0.489
Local complications	18 (11.8%)	15 (7.17%)	*p* = 0.092
Need for antibiotics	64 (42.1%)	57 (27.2%)	*p* = 0.014
Duration of hospitalisation	6.4 ± 4.7	4.2 ± 3.5	*p* = 0.003

**Table 4 medicina-59-00070-t004:** Categorisation according to Prognostic Nutritional Index (PNI).

	Low-PNI < 45 (N:210)	High-PNI ≥ 45 (N:151)	*p* Value
Severe pancreatitis	66 (31.4%)	2 (1.32%)	*p* < 0.001
Hospitalisation in intensive care unit	24 (15.9%)	0	*p* < 0.001
Exitus	8 (3.8%)	0	*p* < 0.001
Need for surgery	18 (11.9%)	0	*p* < 0.001
Local complications	33 (15.7%)	0	*p* < 0.001
Need for antibiotics	101 (%48)	20 (13.2%)	*p* < 0.05
Duration of hospitalisation	5.04 ± 3.8	4.72 ± 3.14	*p* < 0.05

**Table 5 medicina-59-00070-t005:** Relationship between PNI and CONUT scores.

	Low-CONUT (≤2)	High-CONUT (≥3)
low-PNI (<45)	80 (38.1%)	130 (61.9%)
high-PNI (≥45)	129 (85.4%)	22 (14.6%)

## Data Availability

The data presented in this study are available on request from the corresponding author.

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
