# Peer review of "Controlling Nutritional Status (CONUT) Score and Prognostic Nutritional Index (PNI) Are Good Candidates for Prognostic Markers for Acute Pancreatitis"

_medicina, 2022, doi:10.3390/medicina59010070_

Round 1
Reviewer 1 Report
To the authors
In this manuscript, the authors presented that CONUT and PNI are useful prognostic markers for estimating severity and results of AP. However, I advise the authors to make some modifications in the manuscript.
Major points
#1. When did the authors evaluate CONUT and PNI? At the diagnosis of AP? CONUT and PNI are affected by inflammation, so the timing of making evaluation of CONUT and PNI are important. Please describe in the Materials and Methods section.
#2. I want to request the authors to make a table that presenting baseline characteristics of each group of patients. Did the inflammation markers such as WBC or CRP are equivalents at the diagnosis of AP? How about BMI or severity of AP? Also, I want to request the authors to show the results of each parameter of CONUT and PNI in the both groups.
#3. I want to know the difference of CONUT and PNI. Please describe in the text.
#4. What kind of surgery was needed? Please describe.
#5. Why these nutritional parameters affect the treatment course? Please describe in the Discussion section more detail, if possible.
Author Response
Dear;
Editor of the Nutrients
Manuscript reference number: medicina-2087090
First of all, thank you very much for evaluating our study and subjecting it to reviewers' criticism. Our manuscript has improved further with your criticisms and suggestions. Desired changes and additions were made to the article, taking into account the suggestions of the reviewers. The sections where changes were made are shown in red.
For this;
- The evaluation time of the PNI and CONUT score has been added to the Methods section.
- A table containing patient characteristics was prepared. The role of WBC and CRP in predicting prognosis in acute pancreatitis is limited. Although there are many scoring systems such as Ranson, APACHE 2, SIRS, BISAP, BALTAZAR to determine the severity of pancreatitis, the comparison was made according to the ATLANTA classification as the best indicator was the ATLANTA classification. As our study was retrospective, we do not know the BMI of the patients. Therefore, we cannot make a comment between BMI and AP severity.
- The difference between the CONUT score and the PNI is noted in the text.
- The reason why the surgery was needed and what was done were stated in the method section.
- Why these nutritional parameters might affect the treatment process has been added to the discussion section.
Reviewer 2 Report
I would like to congratulate the authors on their work! This is potentially significant research since it provides evidence for the potential use of PNI and CONUT as a prognostic marker in acute pancreatitis (AP).
However, there are some aspects that must be addressed.
1. English needs to be improved and typo errors corrected.
Examples:
Line 10: beneficial for assessment -> beneficial for the assessment
Line 18: had moderate / severe AP -> had moderate/severe AP
Line 21: no differences with regard to surgical -> no differences concerning surgical
Line 24: estimating severity -> estimating the severity
Line 29: inflammation of pancreas -> inflammation of the pancreas
Line 30: due to gland's auto-digestion -> due to the gland's auto-digestion
Line 39: To determine severity -> To determine the severity
Line 41: an APACHE score ≤ 9 and with their Ranson score -> an APACHE score ≤ 9, and their Ranson score
Line 42: were considered having -> were considered to have
Line 57: a negative impact on both on nutritional status and progression of the disease -> a negative impact on both nutritional status and the progression of the disease
Line 67: PNI and CONUT as a prognostic marker -> PNI and CONUT as prognostic markers
And so on… all over the manuscript.
2. Line 202: the font format is different in this line.
3. Review the manuscript and delete the spaces between values and letters/words. Example: “Low <45 (N: 210)” -> “Low<45 (N:120)”.
4. Please keep the same format of “p-value” all over the manuscript, including the tables and figures (see table 2 and table 3, having “P value” and “p value”).
For Table 2: I recommend using “high-CONUT (≥3)” and “low-CONUT (≤2)”.
For Table 3: I recommend using “low-PNI (<45), N:210” and “high-PNI (≥45), N:151”
For Table 4: I recommend using “low-PNI”, “high-PNI”, “low-CONUT” and “high-CONUT”.
Please be consistent with the format and formulations all over the manuscript.
5. Rename the “Findings” section into “Results”.
For Results section: Avoid repeating the same information that is stated in the Methods section and in the tables. Just present the important outcomes, not all the percentages of patients and diseases. Also, avoid repeating “PNI” and “CONUT” for so many times in the same paragraph.
Moreover, I suggest including the ROC analyze, and identify the optimal cut-off value of the PNI, and make the multivariate analyze to identify the predictors of severe pancreatitis and maybe ICU admission. In my opinion is very important to identify the cut-off value of nutritional markers regarding the poor outcome.
6. The Discussion section is lacking information. In this section, the authors should compare their results to other researchers. Don’t repeat again the results and don’t use information that is already in the Introduction section.
And because markers are a very interesting topic that has received a lot of attention in the latest years, I strongly advise the authors to improve the quality of the research by comparing the results with articles (5-7) that are new (from 2021-2022) in the Discussion section. Currently, most references are over 5-10 years old. See the following list as examples:
- https://doi.org/10.3390/diagnostics12112757
- https://doi.org/10.3390/nu14204433
- https://doi.org/10.3390/life12091447
-https://doi.org/10.3390/cancers14205075
- https://doi.org/10.3390/nu14112317
- https://doi.org/10.3390/nu14102105
Kind regards
Author Response
Dear;
Editor of the Nutrients
Manuscript reference number: medicina-2087090
First of all, thank you very much for evaluating our study and subjecting it to reviewers' criticism. Our manuscript has improved further with your criticisms and suggestions. Desired changes and additions were made to the article, taking into account the suggestions of the reviewers. The sections where changes were made are shown in red.
For this;
- Spelling, grammar, repetitive statements, and word usage errors were corrected in the text.
- Fixed the font in the line.
- The space between values and letters/words in the text has been deleted.
- The p value in tables and figures were written in the same format.
- The name of the "Findings" section has been changed to "Results". Duplicates were removed in the method and conclusion part. Only percentages were written in the result section. We need additional time to perform ROC analysis.
- Excerpts from current articles were also included in the Discussion section. We used different information from the information in the introduction. To our knowledge, there was no other study using inflammation and nutritional markers such as CONUT score and PNI in acute pancreatitis. Therefore, we could not compare the results with other studies. However, we refer to studies on other subjects.
Round 2
Reviewer 2 Report
No further comments. Well done.